# Functionalisation of Electrospun Cellulose Acetate Membranes with PEDOT and PPy for Electronic Controlled Drug Release

**DOI:** 10.3390/nano13091493

**Published:** 2023-04-27

**Authors:** Beatriz Lago, Miguel Brito, Cristina M. M. Almeida, Isabel Ferreira, Ana Catarina Baptista

**Affiliations:** 1CENIMAT|I3N, Materials Science Department, School of Science and Technology, NOVA University of Lisbon, 2829-516 Caparica, Portugal; 2Laboratory of Bromatology and Water Quality, Faculty of Pharmacy, University of Lisbon, 1649-003 Lisbon, Portugal; 3iMed.UL (Institute for Medicines and Pharmaceutical Sciences, Portugal), Faculty of Pharmacy, University of Lisbon, 1649-003 Lisbon, Portugal

**Keywords:** controlled drug release system, cellulose acetate, electrospinning, conducting polymers, polypyrrole, PEDOT, electrical stimuli

## Abstract

Controlled drug release via electrical stimulation from drug-impregnated fibres was studied using electrospun cellulose acetate (CA) membranes and encapsulated ibuprofen (IBU). This research outlines the influence of polypyrrole (PPy) and poly(3,4-ethylenedioxythiophene) (PEDOT)-functionalised CA membranes and their suitability for dermal electronic-controlled drug release. Micro Raman analysis confirmed polymer functionalisation of CA membranes and drug incorporation. Scanning electron microscopy (SEM) images evidenced the presence of PPy and PEDOT coatings. The kinetic of drug release was analysed, and the passive and active release was compared. In the proposed systems, the drug release is controlled by very low electrical potentials. A potential of −0.3 V applied to membranes showed the ibuprofen retention, and a positive potential of +0.3 V, +0.5 V, or +0.8 V, depending on the conductive polymer and membrane configuration, enhanced the drug release. A small adhesive patch was constructed to validate this system for cutaneous application and verified an “ON/OFF” ibuprofen release pattern from membranes.

## 1. Introduction

Controlled-release drug technology has progressed over the last decades [1]. Recent advances studied the clinical application of intelligent polymers, hydrogels, and biodegradable polymers in controlled long-term release formulations [2]. In this study, we attempted to construct a simple control drug delivery system for dermal applications constituted by cellulose acetate (CA) membranes as drug carriers and CA functionalisation with conductive polymers to allow electrochemical control. Electrospun polycaprolactone (PCL) matrices loaded with antibiotics were tested by Bölgen et al. [3] in 2007 to prevent post-surgical abdominal adhesions in rats. The authors concluded that 80% of the drug previously adsorbed in the matrix was released in only three hours, indicating the lack of efficiency of the adsorption method for high concentrations of the drug. Electrospun cellulose acetate (CA) fibres (20% weight of CA used) were tested by Tungprapa et al. [4] to release four model drugs. As such, nonwoven CA nanofibers have raised enormous interest as a polymeric matrix for immobilising drugs and other bioactive substances, such as vitamins. Some vitamins, topically used in the skin, are powerful therapeutic agents for treating cutaneous pathologies. In this group, vitamin A and vitamin E exhibit multifaceted biological action. Thus, Taepaiboon et al. in 2007 immobilised these two vitamins on electrified CA nanofibers, highlighting the gradual increase in the cumulative release of the vitamins during the release period from the electro-phased CA fibres, in contrast to their immediate release from the corresponding fused CA films [5].

Concerning membranes/fibres, functionalisation is performed to increase functionalities, such as higher strength, higher flexibility, more acid or basic media stability, and better optical, electrical, or magnetic properties. In the specific case of the application of this work, to develop a system that responds to electrical stimuli, the membrane must be electrically conductive. This is possible by functionalising the membrane with a conducting polymer such as polypyrrole (PPy) or poly(3,4-ethylenedioxythiophene) (PEDOT) [6,7]. Laforgue et al. [8], in 2010, studied the polymerisation of PEDOT in the vapour phase to obtain polyvinylpyrrolidone electrically conductive fibres with the highest conductivity values reported at that time for nanofibers produced by this polymerisation technique. However, in 2014, single-crystalline PEDOT nanowires were produced by this polymerisation technique, achieving conductivities of 8797 S/cm [9]. Müller et al. [10], in 2011, succeeded in coating bacterial cellulose membranes with PPy via in situ pyrrole polymerisation. It consists of immersing a membrane in an aqueous solution of pyrrole, which is then polymerised by adding a catalyst, iron (III) chloride (FeCl_3_). Tang et al. [11], in 2015, produced flexible and conductive composite membranes by in situ chemical synthesis of pyrrole in the presence of bacterial cellulose and subsequently treated the membrane with a polysiloxane solution. The results showed a continuous structure constituted by the PPy particles deposited on the cellulose surface. Baptista et al. [12], in 2018, developed bio-batteries based on flexible, lightweight, and biocompatible material using conductive cellulose acetate-based electrospun fibres functionalised with PPy, through in situ polymerisation. Carli et al. [13] studied, for the first time, in 2019, the neuroprotective and anti-inflammatory properties of tauroursodesoxycholic acid (TUDCA). This natural bile acid was incorporated into PEDOT, and the new material, PEDOT-TUDCA, efficiently promoted an electrochemically controlled drug release while preserving the electrochemical properties. Furthermore, the low cytotoxicity observed with viability tests makes PEDOT-TUDCA an excellent candidate to prolong the period of chronic neural recording in brain implants.

Although some functional systems have been developed in recent years regarding the controlled release of drugs from electrical stimuli, they are yet to be applied to humans. Electro-responsive hydrogels, conductive polymers, and layer-by-layer electro-responsive films are the main routes studied for the controlled electronic release of drugs [14]. According to Giuseppi-Elie [15], electronically controlled drug release devices produce a programmed release profile influenced by the application of voltage or even current. In 2006, Wadhwa et al. [16] developed one of the first drug release systems whose control is performed electronically. This consists of two electrodes, on which a PPy film is deposited by electropolymerisation of pyrrole, containing dexamethasone (anti-inflammatory) that works as a dopant due to its negative charge of the conductive polymer. The release of the drug was performed using the cyclic voltammetry technique, and it was verified that the reduction and oxidation peaks characteristic of pyrrole only appeared from the third scanning cycle and that the greater the number of cycles, the more intense the peaks. Abidian et al. [17] studied the controlled release of dexamethasone by applying external electrical stimuli on PEDOT nanotubes. Biodegradable poly(L-lactide) (PLLA) or poly(lactic-co-glycolic acid) (PLGA) nanofibers were electrophoresed and conductive polymers around the electrophoresed nanofibers were deposited by an electrochemical process. Applying +1 V between a platinum electrode and the nanotubes loaded with 2 mg of the drug allowed a linear control of the drug release for up to 200 h for a release of 1.5 mg. In 2013, Servant et al. [18] tested PMAA-pMWNT (polymethacrylic acid—multi-walled carbon nanotube) hydrogels in the release of radioactively labelled sucrose, which in turn gave rise to a pulsatile release profile of 14C-sucrose, encapsulated in the polymer matrix by swelling, when “ON/OFF” pulses of direct current electric field (+10 V for 15 min) were applied. It was concluded that about 70% was released after 80 min exposure to the electric field. In 2014, Krukiewicz et al. [19] successfully developed a PEDOT-based system for the electronically controlled release of ibuprofen in ionic form. The electrochemical synthesis was optimised to obtain a conductive matrix with the highest possible drug content. The drug release process by electrical stimuli was studied by applying stimuli between −0.8 V and +0.8 V, with a silver counter electrode, which showed that the highest amount of released drug occurred when the stimulus of −0.5 V was applied. The concentration of released ibuprofen was shown to be dependent on the redox state of the polymer matrix. Therefore, the application of negative potential resulted in the release of the drug, while positive potential resulted in the retention of the drug. In 2017, Samanta et al. [20] developed a re-absorbable and electro-responsive drug release system that includes nanocomposite films of a methylmethacrylate-co-methacrylic acid-based polymer to control the released drug dose. The authors concluded that it evolves linear dependency on the applied potential. In 2019, Feiner et al. [21] developed a multifunctional device capable of performing the controlled release of different drugs and simultaneously monitoring the applied current between a cathode, covered with PPy containing the drug, and a silver anode. The authors demonstrated that the relationship between the applied current, the content encapsulated in the fibrous layer of PPy, and the amount of drug released (dexamethasone, indomethacin, and aspirin) is linear. In 2021, Baptista et al. [22] studied the electronically controlled release of ibuprofen (IBU) previously impregnated in gauze and CA membranes. They obtained a conductivity range of 1–10 mS/cm in gauze functionalised with PPy and CA fibres, providing a controlled drug release by electrical stimuli in a system composed of PPy/IBU/PPy membranes and a silver electrode. A small dermal patch constructed with these membranes retained ibuprofen at +1.5 V and released it rapidly at −0.5 V.

Over the last few years, a significant evolution in developing new drug release systems for dermal applications has occurred [1,23,24,25]. Currently, there are already devices that react to mechanical stimuli [26], are electrically flexible [27], have iontophoresis under the skin [28], and are multifunctionally controlled by temperature [29] and electrical stimuli. Hydrogel nanocomposites represent another family of trans-dermal drug release systems that have been controlled by temperature [30], pH [31], electrical stimulation [32] or are self-powered [33] and used for wound healing [31]. Microchips, capsules, pumps, self-injectables, and transdermal patches are examples of commercially available electronically controlled drug delivery systems [34]. While there are significant advantages to using these systems, much work must be done to make them easier to use and improve patient compliance. In this field, regular topical administration of drugs for skin disease is one of the primary needs [35].

The present work aims to develop an electrically stimulated drug delivery textile for precise and on-demand drug dosing envisaging dermal applications. The active transdermal drug delivery system reported here relies on a non-invasive drug-releasing mechanism, with the possibility of adjusting drug release dosing by the application of an external electrical stimulus. This work proposes the production of an electrospun IBU containing CA fibres followed by membrane functionalisation with PPy and PEDOT. Ibuprofen is a nonsteroidal anti-inflammatory drug (NSAID) often used as a model to study transdermal drug delivery systems due to its ability to efficiently diffuse through the epidermal and dermal layers [36,37]. PPy_90_/CA (cellulose acetate membrane functionalised with PPy for 90 min) and PEDOT_120_/CA (cellulose acetate membrane functionalised with PEDOT for 120 min) composite membranes were electrochemically evaluated under simulated physiological conditions and assembled into five different systems to build a patch that can release a specific drug amount in response to a specific electric stimulus.

## 2. Materials and Methods

### 2.1. Production of Membranes

The cellulose acetate membranes were produced using a polymeric solution of 12% wt cellulose acetate (Mn 50,000 with 40% acetyl groups, Sigma Aldrich, Stainheim, Germany) in a solvent system of acetone 99.5% (Honeywell Riedel-de Haën, Seelze, Germany) and dimethylacetamide (DMAc) (Carlo Erba Reagents S.A.S., Val de Reuil, France), in a ratio of 2:1, respectively. Additionally, ibuprofen was added to a CA solution to produce CA fibres with ibuprofen (CA:IBU): 1 mg of ibuprofen 99.3% (FarmaQuimica Sur S.L., Málaga, Spain) was added to the 2 g solution of CA. For the electrospinning (ES) process, the homogenised solution was added to a syringe (1 mL from Injekt, B. Braun, Melsungen, Germany) with a metallic needle of 21 gauge (internal diameter of 4.5 mm from ITEC, Iberiana Technical, Braga, Portugal) and placed in an infusion syringe pump (model NE300 from New ERA Pump Systems, Farmingdale, NY, USA) to control the flow rate (0.20 mL/h) that the polymeric solution was ejected during ES process. A high voltage source (incorporated in a modular robotic system, LRC MultiCoater, Oslo, Norway) was used to apply a high voltage (20 kV) to the metallic tip, and a grounded static collector (metallic foil) was placed 15 cm from the needle. The process schematic is presented in Figure 1a. Finally, to obtain a similar IBU amount in all sample portions, the electrospun membrane was cut into 16 concentric triangles.

### 2.2. In Situ Oxidation of Pyrrole in Aqueous Solution

A CA or CA:IBU membrane was immersed in an aqueous solution of 0.05 mol/L pyrrole (98%, Sigma-Aldrich, Stainheim, Germany) for 10 min to allow pyrrole impregnation. After this period, an oxidising solution of iron (III) chloride hexahydrate (FeCl_3_·6H_2_O, purity 99%, Chem-Lab NV, Zedelgem, Belgium)—in a mass ratio of 2:1 (FeCl_3_:pyrrole) [12]—was gently added. The pyrrole polymerisation was carried out for 90 min, and after that time, the membranes were carefully removed from the solution and washed with ultrapure water (Elix^®^ purification system, Merck Millipore) and ethanol (Honeywell Riedel-de Haën, Seelze, Germany, 99.5%) to extract the by-products and unreacted residues. Finally, membranes were dried at room temperature. This functionalisation process is illustrated in Figure 1b.

### 2.3. Vapour Phase Polymerisation of EDOT

The PEDOT functionalisation of the membranes through exposure to EDOT vapours was adapted from the literature [38]. The electrospinning membranes were immersed in an aqueous solution of 40 g/L of an oxidising agent, hexahydrated iron (III) chloride (FeCl_3_·6H_2_O, purity = 99%, Chem-Lab NV, Zedelgem, Belgium) for 20 min under 100 rpm stirring. The membranes impregnated with the oxidising agent were carefully removed from the solution and left to dry at room temperature. Once dried, they were fixed inside a closed chamber with 0.5 mL of EDOT monomer (Sigma-Aldrich, Stainheim, Germany, 97%) at the bottom of the container. The closed chamber was introduced in an oven (Memmert, Schwabach, Germany) at 90 °C for 120 min to initiate EDOT polymerisation, as illustrated in Figure 1c. After this polymerisation process, the coated membranes were removed from the chamber and washed with ultrapure water (Elix^®^ purification system, Merck Millipore, Darmstadt, Germany) and abundant ethanol 99.5% (Honeywell Riedel-de Haën, Seelze, Germany) to remove un-reacted residues and by-products. 

### 2.4. Characterisation

The morphology of samples was analysed by scanning electron microscopy (SEM) (model Hitachi S2400, Tokyo, Japan). The samples were fixed on a metal sample holder using a double-sided carbon-based conductive tape (Electron Microscopy Sciences) and then coated with a thin layer of gold–palladium alloy. From the higher resolution images obtained, the mean diameter of fibres was estimated using an image processing software (ImageJ^®^, NIST, Gaithersburg, MD, USA), considering 30 different measurements for each sample.

A Confocal Raman Spectrophotometer (Witec Alpha 300 RAS, Ulm, Germany) using a 532 nm laser and a power of 0.5 mW was used to evaluate the chemical composition of the membranes.

### 2.5. Conductivity

The electrical conductivity was obtained by measuring the current–voltage (I–V) values using a pico-amperemeter/voltage source (model 6400, Keithley Instruments, Germering, Germany), two probe tips (Alessi REL-450, Bromont, Canada), and a computer for recording data. The thickness of each membrane was measured with a micrometre in 15 different areas of the membrane, and then calculated the average thickness. The I–V measurements were repeated three times for each sample, and the average of the slopes was obtained for each membrane. The planar conductivity was measured on the surface of the membrane according to Equation (1),
(1)$$\sigma =\frac{l}{AR}$$
where *σ* is the electrical conductivity (S cm^−1^), *R* is the resistance (Ω) obtained from the slope of the linear I–V plot, *A* the area of the cross-section of the membrane (cm^2^), and *l* is the distance between electrodes (cm).

A conductive carbon paint (Bare Conductive^®^, Barcelona, Spain) was used to obtain the electrodes with a rectangular shape of dimensions 5 mm by 3 mm, 1 mm apart. The probes were placed on the contacts, and the I–V values were acquired. In this case, the cross-section was delimited by the electrode length and the thickness of the membrane. The transverse conductivity was measured along the thickness of the membrane. For that, the membrane was pressed between two aluminium conductive tape dimensions 6 mm by 6 mm, concerning the area in contact with the membrane corresponding to the cross-section area of the measured membrane.

### 2.6. Cyclic Voltammetry

Cyclic voltammetry was carried out for various membrane systems using a Gamry Instruments potentiostat (Reference 3000). A two-electrode configuration was used for cyclic voltammetry measurements. In this configuration, the working electrode corresponds to the sample we are testing, and the counter electrode is connected to the silver wire (1 cm apart), which completes the circuit and maintains a constant interfacial potential. The potentials and the current between both electrodes were measured. This configuration was commonly used for solid-state electrochemistry [12,39,40]. Five different membrane systems were analysed: (A) PPy_90_ Rim System, consisting of two identical CA membranes polymerised with PPy for 90 min, with a CA:IBU membrane in between. An insulating adhesive tape (Kapton^®^ tape, DuPont, Circleville, OH, USA) was used around the membranes systems to seal the borders of the device and ensure that release does not occur from the “edges”. (B) PPy_90_ Single System, consisting of a CA:IBU membrane polymerised with PPy for 90 min. (C) PEDOT_120_ Rim System, consisting of two identical CA membranes polymerised with PEDOT for 120 min, with a CA:IBU membrane in between. This system was sealed with adhesive tape (Kapton^®^ tape, DuPont, Circleville, OH, USA). (D) PEDOT_120_ Single System, consisting of a CA:IBU membrane polymerised with PEDOT for 120 min. (E) Mix Rim System, consisting of two identical CA membranes, one polymerised with PPy for 90 min and the other with PEDOT for 120 min, with a CA:IBU membrane in between. This system was sealed with adhesive tape (Kapton^®^ tape, DuPont, Circleville, OH, USA). All the measurements were performed by adding 20 mL of a Simulated Body Fluid (SBF). The preparation of this solution was based on the procedure described by Kokubo et al. [41].

### 2.7. Drug Release

Release tests were performed in SBF. The membrane systems were placed in a container with 20 mL of SBF, with the membrane 1 cm away from an Ag wire. Different voltage potentials were applied using a voltage source corresponding to continuous electrical stimuli between the membranes and a silver electrode. The membrane was connected to the positive terminal of the voltage source, while Ag was connected to the ground electrode. A voltage was applied during a specific period and repeated several times to define the amount of drug released along the time. A passive drug release profile was also studied for each membrane system. 

All tests considered an initial period of diffusion of 1 min, followed by a 1 min period at a fixed electrical stimulation voltage for 10 min. The tests were performed cumulatively, i.e., the electrical stimulation was applied for 1 min, then switched off, and a sample of the release medium was taken and placed in a 3 mL quartz cell to perform the absorbance spectra. The extracted medium returned to the release vessel, the membrane was again immersed, and the fixed voltage was then applied again.

The UV-Visible spectroscopy was performed after each stimulus to determine the concentration of the drug released to the medium using a calibration curve of ibuprofen in SBF [42]. The SBF solution was used as a reference. The wavelength scan of IBU and its calibration curve are shown in Appendix A, respectively.

The confirmation of ibuprofen release from polymer matrices was performed by ultra-performance liquid chromatography–tandem mass spectrometry (UPLC-MS/MS) using a Dionex Ultimate 3000 system (Thermo Fisher Scientific, Massachusetts, USA) coupled to a mass spectrometer TSQ Endura triple quadrupole from Thermo Scientific and a Kinetex EVO C18 column (2.1 cm × 50 mm × 2.6 μm) from Phenomenex (Torrance, CA, USA). The tandem mass spectrometer operated in positive ion ESI mode using multiple reaction monitoring (MRM) mode. The chromatographic run uses a mobile phase A with a mixture of H_2_O + 0.01 mM ammonium acetate (NH_4_Ac) + 0.5% formic acid (HCOOH) (*v*/*v*) and mobile phase B with 100% methanol (MeOH). The gradient program started with 95% mobile phase A, followed by a linear decrease to 50% until 3.0 min, 30% until 5.5 min, and 10% until 8.0 min (held 3.0 min). To re-equilibrate the system, an increase of mobile phase A to 95% was performed in 1.0 min (held 3.0 min). The injection volume was 20 µL, and the flow rate was 0.3 mL/min. Triple quadrupole operating conditions were optimised for multiple reaction monitoring mode (MRM). The optimised MS/MS conditions were based on the selection of ionisation mode, optimum collision energy (V), ion transfer tube, vaporiser temperatures (°C), and accurate radio frequency (RF) lens for each compound. Nitrogen was used as a sheath, auxiliary, and sweep gas. Sheath gas flow was set to 40 Arb, and the aux gas flow was set to 10. Ion transfer tube and vaporiser temperatures were also optimised, with the selected temperatures being 300 °C for the ion transfer tube and 200 °C for the vaporiser (Appendix A).

After optimisation of the triple quadrupole conditions, only one transition (product ion) was established, the transition for quantification (MRM1). The optimisation and selection of the transition conditions were carried out by injecting the individual standard of IBU with a concentration around 2 mg/L in the Rheodyne valve of the MS (without chromatographic separation). The optimised mass spectrometer conditions are summarised in Appendix A.

### 2.8. Prototype Drug Release

The prototype developed contains a voltage divider consisting of a resistor and a potentiometer connected to an external battery and a silver wire, the respective membrane systems, conductive tape, and a wound dressing. The external battery used was a commercial 1.5 V button cell. Since the voltages studied are relatively low, resistors were included in the prototype. In addition, a potentiometer was used to adjust the resistance to each release system and interchange between two different voltages in a single circuit. To study the release with the prototype, the electric circuit was assembled and connected to the membrane system chosen for testing through a crocodile attached to the conductive tape glued to the upper part of the membrane (Figure 2). A wound adhesive dressing was placed on top of the membrane to simulate possible contact with the skin. A 0.1 mm diameter silver wire was glued to the underside of the dressing adhesive. During the entire release, a multimeter was connected to the circuit to ensure that the voltages to be applied were correct and did not change during the study.

## 3. Results and Discussion

### 3.1. Membranes Characteristics

The successful encapsulation of ibuprofen (IBU) and the functionalisation of cellulose acetate (CA) membranes with polypyrrole (PPy) and poly(3,4-ethylenedioxythiophene) (PEDOT) were evaluated by Raman spectroscopy. Figure 3a shows the spectra obtained from the individual constituents of the membranes in their original state, namely, PEDOT, PPy, CA and IBU. By analysing these spectra, it is possible to identify the characteristic peaks of each compound, which allow their further identification in the functionalised membranes. Figure 3b shows the spectra for membranes PPy_90_/CA:IBU and PPy_90_/CA and cellulose acetate membrane with ibuprofen (CA:IBU). Figure 3c shows the spectra of PEDOT functionalised membranes, namely PEDOT_120_/CA:IBU and PEDOT_120_/CA, CA:IBU membrane is included for comparison.

Figure 3b,c clearly show the spectral signature of new functional groups in cellulose acetate membranes, mainly those highlighted in the red circles. As previously reported [43], the characteristic peaks of PPy are located at 928 cm^−1^, 971 cm^−1^, 1048 cm^−1^, 1365 cm^−1^, and 1584 cm^−1^. The first two peaks are weak and correspond to deformation vibrations of the glucose ring, the peak at 1048 cm^−1^ is due to simple deformation in the C-H plane, and the last two peaks can be attributed to the ring and C=C bond elongation and stretching vibration modes. In Figure 3b, the peaks 971 cm^−1^, 1048 cm^−1^, 1365 cm^−1^, and 1584 cm^−1^ were identified, which evidences the functionalisation of CA with PPy, i.e., PPy coats the surface of the membrane fibres.

Regarding the characteristic peaks of PEDOT, these were located at 1270 cm^−1^, 1380 cm^−1^, 1451 cm^−1^, and 1519 cm^−1^ [44], as observed in the PEDOT spectrum. These peaks correspond to the C-C bond stretching, C-C bond stretching, symmetric C=C bond stretching, and asymmetric C=C bond stretching vibration modes, respectively. In Figure 3c, the peaks 1451 cm^−1^ and 1519 cm^−1^ were consistent with the PEDOT functionalisation of membranes.

The IBU reference spectra reveal peaks at 2869 cm^−1^, 2957 cm^−1^, and 3047 cm ^−1^, corresponding to a band in the region 2850–3100 cm^−1^ [45]. These peaks are ascribed to the stretching of the C-H bond, the symmetric stretching of the CH_3_ bond, and the antisymmetric stretching of the H-C-C-H bond, respectively [46]. As indicated in Figure 3b,c, this band is prominent in the Raman spectrum of the PEDOT_120_/CA:IBU and CA:IBU membranes, evidencing the drug’s incorporation. It should be noted that in the PPy_90_/CA:IBU membrane, the presence of IBU is not as evident as in the conductive polymer since the amount of drug present in the membranes is relatively small in proportion with the amounts of the other materials. However, in the region of the band of the peaks characteristic of IBU, it is possible to identify a higher intensity of the spectrum compared to the same region of the spectrum of the PPy_90_/CA membrane without IBU. This indicates that the compound may be present but not in significant quantities that allow its identification. The same happens in the PEDOT functionalisation spectra. However, in the PEDOT_120_/CA:IBU spectrum, it is possible to identify the peaks at 2869 cm^−1^ and 2957 cm^−1^, although with reduced intensity. Compared to the CA:IBU spectrum, a peak at 2957 cm^−1^ was identified for the presence of IBU. Since the drug concentration was low relative to the amount of polymer, it is challenging to detect by this Raman technique.

### 3.2. Morphology

Figure 4 compares fibre morphology obtained by SEM of CA membranes, CA membranes functionalised with PPy, and PEDOT, all with and without IBU. It is observed that both membranes of CA (Figure 4a) and CA:IBU (Figure 4b) were composed of randomly orientated fibres with a high average diameter distribution. The presence of IBU is not detected in the surface of fibres; however, fibres containing IBU showed an increased average diameter compared to those without the drug. This difference can be explained by the increase in solution viscosity with the addition of the drug, leading to the formation of fibres with larger diameters, corresponding to an increase of about 30% in the average diameter of the fibres.

Figure 4 reveals some melted fibres that can be justified by a change in the ideal humidity and temperature parameters during the electrospinning process, which can affect the solution viscosity, but mainly cause a change in the solvent evaporation rate, thus leading to the formation of non-uniform and defective fibres. The SEM image of PPy_90_/AC (Figure 4c) clearly shows a layer of PPy along the fibres’ whole surface, even observing this polymer’s agglomerates. Considering that the polymerisation stands for 90 min, a complete coating of the fibres by the PPy was expected. In PPy_90_/CA:IBU membranes (Figure 4d), the obtained PPy coating is similar, which means that if IBU is released during the membrane polymerisation process, it does not inhibit or affect the polymerisation. For the PEDOT-functionalised membranes (Figure 4e,f), the obtained coating was smoother and more uniform when compared with PPy-coated membranes, with no evidence of PEDOT aggregation on the surface of fibres. The presence of PEDOT coating was also confirmed by Raman spectroscopy, as shown in Figure 3c.

### 3.3. Cyclic Voltammetry

A cyclic voltammetry study was conducted to analyse the potential of oxidation and/or reduction of the selected materials and find the suitable potential window for the electrical stimulation. The influence of the scan rate on the electrochemical behaviour is shown in Appendix A. The dependency of the peak current and peak potential on the scan rate gives information about the reaction mechanism and kinetics involved: at high scan rates, fast kinetic processes can be detected, while at low scan rates, slow processes can be observed. These results indicate a tendency to observe the increase of peak current with the increasing of the scan rate, which suggests that the reaction can be mainly controlled by the diffusion of the electroactive species (i.e., it is not confined to the electrode surface), apart from PPy_90_ Single System that was not conclusive.

Figure 5 compares the voltammograms obtained at 25 mV/s for the sketched configurations. For the PPy_90_ Rim System, two redox couples at +0.05 V/−0.1 V and at +0.35 V/−0.3 V can be clearly identified and are probably associated with the oxidation/reduction states of PPy to polaron and bipolaron states [12,47]. However, for the PPy_90_ Single System, at a scan rate of 25 mV/s, the presence of redox reactions is not evident. Concerning the PEDOT_120_ Rim System, the respective voltammogram obtained at 25 mV/s reveals a large anodic band around +0.4 V, which can be explained by two single anodic peaks superimposed and two cathodic peaks (−0.25 V and −0.5 V) presenting a lower current density when compared with the PPy_90_ Rim System (0.3 mA/cm^2^). These reactions could be associated with the transition oxidation/reduction states of PEDOT. In the Mix Rim System, at 25 mV/s, the oxidation and reduction peaks are well defined with a higher current density in comparison with the previous systems (0.7 mA/cm^2^) with two anodic peaks at +0.2 V and +0.8 V, and two cathodic peaks at −0.5 V and −0.2 V.

The induced electrical drug release needs conductive polymers coating the membranes to repel the drug or retain it according to the potential of the redox reactions of the polymers. So, it is expected that the peak voltages with the highest current values in the previous voltammograms would release more drugs than those with a lower associated current density.

### 3.4. Drug Release Tests

To monitor the amount of drug released during the controlled release tests, the IBU in SBF was quantified through UV/vis spectroscopy. The passive release consisted of drug release by diffusion, i.e., taking advantage of the concentration gradient, to be compared with the active drug release tests, i.e., with the application of a specific voltage between the electrodes. The applied voltages were selected from the results obtained from the cyclic voltammetry studies. Based on these results, some potentials were evaluated (Appendix A), and two voltage values were selected with respect to their suitability to stimulate or retain the drug release. Drug release tests were performed for the five previously mentioned system configurations, PPy_90_ Rim System, PPy_90_ Single System, PEDOT_120_ Rim System, PEDOT_120_ Single System, and Mix Rim System. To compare, exclusively, the influence of electrical stimulus, the initial minute corresponding to the diffusion was subtracted from the active release. However, it was verified that the different systems in contact with SBF immediately undergo diffusional release. In this work, the electrical stimulus was always applied after 1 min of diffusion in SBF for about 10 min in cycles of 1 min. Figure 6 shows the concentration determined from the absorption peak obtained after 1 min of an applied voltage, obtaining the IBU concentration from the calibration curve of Appendix A.

According to Figure 5, the voltammogram obtained for PPy_90_ Single System did not reveal evident redox peaks; thus, we have tested the potentials of −1.5 V, −0.3 V, +0.3 V, +0.8 V, and +1.2 V to evaluate the possible influence of positive and negative voltage values on the drug release.

Figure 6a shows the drug release profiles for the different polarisation values. The profile obtained by applying negative voltages, −1.5 V and −0.3 V, shows lower IBU concentration values than those obtained by diffusion (0 V), which indicates that the drug release was probably delayed by the PPy coating. However, when applying a positive stimulus, the release profile was similar to a passive release (diffusion), with a slight increase in the IBU concentration released when +0.3 V was applied.

As Jin Qu et al. [48] reported, conductive polymers can bind and release different drugs due to their oxidation and partial reduction by electrical stimulation. He found that the release of negatively charged drugs—such as dexamethasone and indomethacin—resulted from the partial reduction of the positively charged aniline trimer that induced the drug molecules to move to the oppositely charged electrode. IBU is a negatively charged molecule, so it is expected to bind electrostatically to the PPy skeleton. A possible mechanism that explains the previous results is that the therapeutic molecule may be repelled from the CA:IBU-coated PPy membrane and attracted to the silver counter electrode for specific potentials according to the charge of the PPy skeleton. To proceed with this study, the voltages of −0.3 V and +0.3 V were selected to evaluate the “ON/OFF” pattern.

Since the PPy_90_ Rim System presents two oxidation/reduction states, according to cyclic voltammetry results, we studied the influence of the voltages of −0.3, +0.3, +0.5, and +1 V on the IBU release profiles. The selected values of −0.3 and +0.3 V correspond to the redox potentials, and the higher values of +0.5 and +1 V were intended to evaluate a regime of voltages that guarantee the full oxidisation of PPy molecules (Figure 6b). It was possible to conclude that for +0.5 V, a clear increase in the IBU profile release was observed compared to other studied voltages and passive release. This result is in accordance with the oxidation peak potential value deviation observed for higher scan rates (Appendix A). Accordingly, to study the “ON/OFF” pattern, the voltages of −0.3 V and +0.5 V were selected.

The release profiles obtained for the PEDOT_120_ Single System are depicted in Figure 6c, and the selected voltages correspond to the oxidation peaks found in the voltammograms, +0.3 and +0.8 V, and finally, −0.3 V was applied to be comparable with other systems. Higher voltage values were not studied since it was verified that the PPy_90_ Single System is not sensitive to voltages above +0.9 V, disabling the control of drug release. This study revealed that −0.3 V retains the IBU release when compared with positive voltages and passive release, as previously observed for the PPy_90_ Single System. Additionally, it was also observed that +0.8 V slightly increased the amount of therapeutics released to the medium after 10 min.

Concerning the PEDOT_120_ Rim System, illustrated in Figure 6d, an electrical stimulus of −0.3 V was used for comparison with the previous systems since it may retain the drug release. On the other hand, the influence of the voltage values of +0.3 V and +0.5 V was studied accordingly to the presence of a large anodic band within this potential window observed in Figure 5. It is possible to conclude that for +0.5 V, there was a clear increase in the release rate of IBU when compared with other voltages and diffusion profiles, and −0.3 V is also likely to decrease or retain the IBU release as expected. For that reason, for the “ON/OFF” pattern study, the voltages of −0.3 V and +0.5 V were selected.

Figure 6e depicts the results obtained for the Mix Rim System for the voltages of −0.3, +0.3, +0.8, and +1.2 V according to the main reason previously stated. However, the results were inconclusive, suggesting a slight drug retention obtained with the voltage of −0.3 and +0.3 V. Similarly, and to make systems comparable, the voltages of −0.3 V and +0.8 V were considered for the “ON/OFF” pattern study. Table 1 summarises the selected voltages for the different systems in “ON/OFF” release studies.

Although a similar behaviour was observed for all systems concerning the voltage values that lead to the highest drug retention, the voltage that maximised the IBU release ranged between +0.3 V, +0.5 V, and +0.8 V for the various systems. This inconsistency in values can be attributed to the possible bending of the systems during immersion in SBF, which can affect the distance between the electrodes [22].

Based on the previous results and given the purpose of the work, we sought the ideal value of the electric potential that would allow a more significant release of the drug by the systems, as well as the most suitable value of the electric potential that, when applied, would prevent the release of the drug totally or significantly into the medium. In other words, the potential would always be applied to the release system, preventing the release of the drug. Only when this potential is OFF would the release occur by applying a potential that stimulated its release. For this reason, to achieve an “ON/OFF” pattern of release, the negative stimulus was applied first, followed by the positive stimulus, alternating between the two for 10 min, 1 min each.

To verify the possibility of building a patch to electronically control the release of the drug into the wound, the “ON/OFF” voltages for each system shown in Table 1 were tested, and the graphs in Figure 7 were obtained. By analysing the results, it is possible to detect the existence of plateaus that demonstrate a halt in drug release that occurs within minutes of the application of the negative IBU retention stimuli. Between each plateau, more drug molecules are being released into the medium in a linear trend associated with applying a positive stimulus.

As expected, the PEDOT functionalised systems showed the ability to release a higher concentration of IBU to the medium compared to the PPy functionalised systems. The Mix Rim System reached a maximum value of IBU concentration between the maximum concentrations reached by the PPy and PEDOT systems.

### 3.5. Prototype Test

The experimental assembly of the prototype is shown in Figure 2a, and the schematic of the voltage divider is in Figure 2b. In this voltage divider, V_in_ corresponds to the voltage source, which in this case, is the 1.5 V battery, R_1_ is a 320 W resistor, V_out_ corresponds to the output voltage being applied to the membrane, and R_2_ refers to the potentiometer, which operates between 0 and 47 W.

Figure 8a,b shows an apparent increase in drug release when a positive voltage is applied, while retention is observed at −0.3 V. The tests reproduced the results previously presented in Figure 7 and demonstrated that the drug release could be controlled by applying a suitable voltage for a specific system.

The amount of drug released into the medium was calculated and compared in percentage with the amount of drug incorporated in the membrane before the release assay. If the mass of IBU (1 mg) is uniformly distributed across the membrane, after dividing it into 16 identical triangles, we would initially have 62.5 µg of IBU/membrane in each membrane. Using the release concentrations reached after 11 min of release, the amount of IBU existing in 20 mL of SBF was determined, and the proportion of released drug was calculated and obtained as drug release percentages in the range of 300–800% depending on the voltage potential (Appendix A). As the spectrophotometric analysis is not selective, any substance released into the medium and absorbed in the same UV region will affect the results, giving rise to release percentages higher than 100%. One hypothesis is the release of PPy or PEDOT residues during the release assays. The residues are most likely polymer particles from PPy aggregates that detached from fibre surfaces during the release tests. This hypothesis was confirmed by diffusion experiments performed with a membrane of PPy_90_/CA and PEDOT_120_/CA, as illustrated in Appendix A. The release percentages were systematically higher on PEDOT membranes, consistent with the enhancement of the absorption peak of each polymer in SBF. Appendix A presents the results of the tests applied for evaluating the linearity and working range of the UPLC-MS/MS method for the concentration range of 0 to 14.5 µg/L ibuprofen. In Appendix A, at 211 nm and after 11 min, the PPy_90_/CA membrane reached an absorbance of 0.35, while the PEDOT_120_/CA membrane reached an absorbance of 0.8. It is thus verified that at the end of the same time interval and for the same wavelength in the visible region, more PEDOT is released into the medium than PPy, which in turn masks the actual amount of IBU released into the medium. Given the lack of selectivity of the spectrophotometric method, some IBU assays were performed by UPLC-MS/MS, which allowed the confirmation of ibuprofen release at concentrations between LOD and 33 µg/L.

Appendix A shows the working range curve used to quantify ibuprofen in the selected membrane samples to confirm ibuprofen release. The results show that the straight calibration line between 0–14.5 µg/L shows a good correlation (R^2^ ≥ 0.995) and good linearity with a coefficient of variation of the method (CVm) and residual analysis below 5.0%.

The provisional quantification limit (10× quotient of the residual standard deviation of the curve and the slope) was lower than the first point of the calibration curve (1.5 µg/L), indicating the concentrations of the various points of the calibration line are well-adjusted. The first concentration level studied, situated above the limit of quantification, was the standard of 1.5 µg/L. Consequently, the operating range was 1.5–14.5 µg/L.

Table 2 shows the concentration and percentage of ibuprofen obtained in the samples analysed and the respective percentage of release. The obtained concentrations correspond to less than 1% of the initial IBU mass incorporated into the membrane (e.g., 62.5 µg IBU/membrane). This low release rate can be related to inadequate homogenisation of the release solution, the possible release of therapeutic molecules during membrane functionalisation (for single systems), or a possible drug entrapment within Rim Systems.

## 4. Conclusions

In this work, different systems for controlled drug release envisaging dermal applications were tested. The model drug selected was ibuprofen, one of the most common anti-inflammatory drugs. The IBU was encapsulated in an electrospun cellulose acetate membrane. Five different configurations were assembled and tested for electrically controlled drug release.

The study of the effect of the applied voltage between the membrane systems and the silver electrode led to the conclusion that the release of the drug was delayed when a negative potential was applied in comparison with the passive release (diffusion). In contrast, rapid release was observed with positive voltages. For the five systems, the most significant drug retention was observed at a potential of −0.3 V, and the voltages that stimulated release were at a potential of +0.3 V for the PPy_90_ Single System, +0.5 V for the PPy_90_ Rim System and PEDOT_120_ Rim System and, finally, +0.8 V for the PEDOT_120_ Single System and Mix Rim System.

Given the lack of selectivity of the spectrophotometric method, some IBU assays were performed by UPLC-MS/MS, which confirmed the release of ibuprofen in percentages below 1%. Using alternating voltages, an ON/OFF release profile was achieved for all systems. These conditions were successfully replicated in a prototype transdermal patch containing the membranes, a thin silver wire, and covered with a clean dressing to simulate possible skin contact.

When fully developed, this system is expected to be a valuable contribution to personalised medicine, taking advantage of its ON/OFF release mechanism that enables the user/patient to precisely control when and how much of a drug is released.

## Figures and Tables

**Figure 1 nanomaterials-13-01493-f001:**
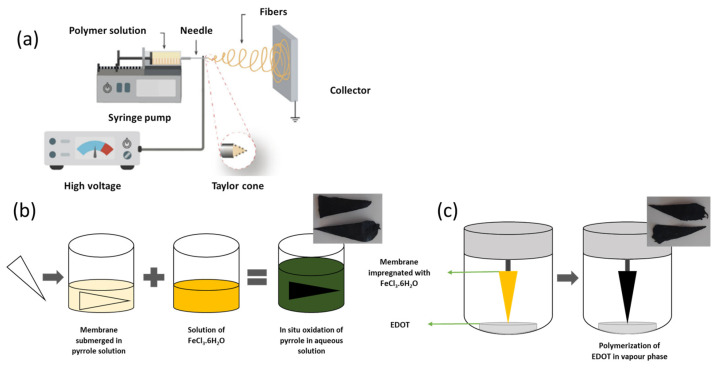
Production and functionalisation of the membranes: (**a**) electrospinning set-up; (**b**) liquid phase in situ polymerisation of pyrrole; (**c**) vapour phase polymerisation of EDOT.

**Figure 2 nanomaterials-13-01493-f002:**
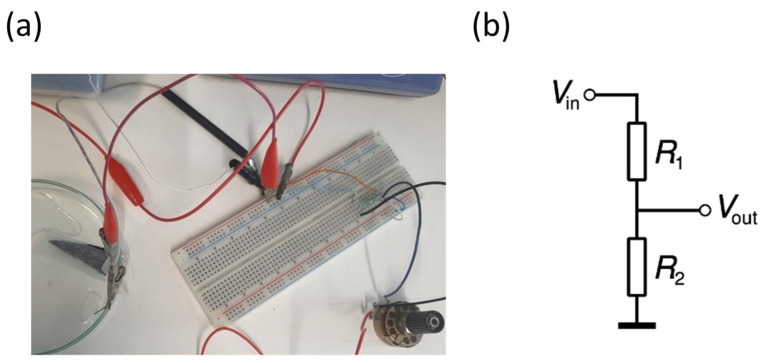
(**a**) Experimental assembly of PPy_90_ Single System prototype and (**b**) Circuit voltage divider.

**Figure 3 nanomaterials-13-01493-f003:**
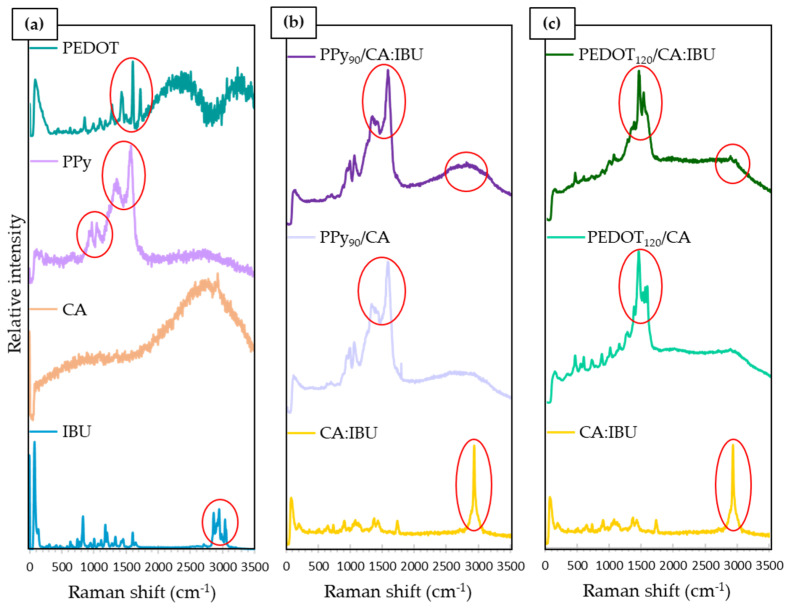
Raman spectra of: (**a**) the single materials, PEDOT, PPy, CA, and IBU; (**b**) PPy_90_/CA:IBU, PPy_90_/CA, and CA:IBU; (**c**) PEDOT_120_/CA:IBU, PEDOT_120_/CA, and CA:IBU. Red circles highlight the characteristic peaks of each compound.

**Figure 4 nanomaterials-13-01493-f004:**
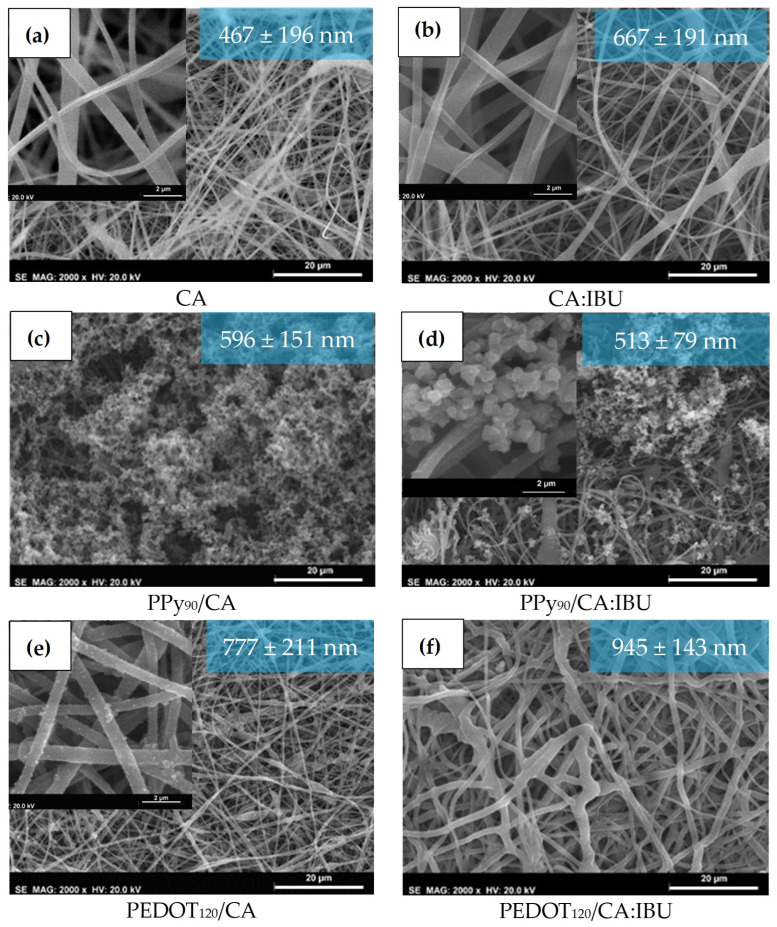
SEM images of CA, PPy_90_/CA and PEDOT_120_/CA membranes without (**a**,**c**,**e**) and with ibuprofen (**b**,**d**,**f**). The average fibre diameter (30 measurements obtained for each sample) and the standard deviation are presented in the blue box.

**Figure 5 nanomaterials-13-01493-f005:**
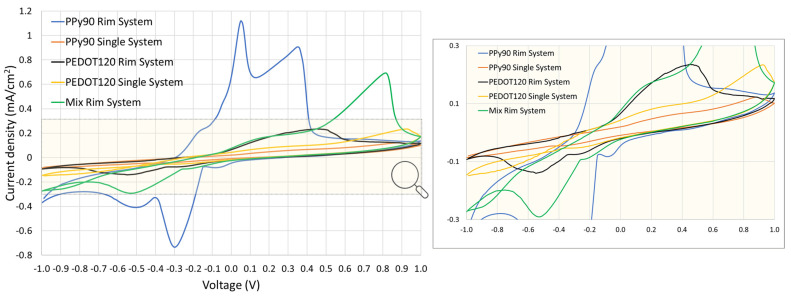
Third cycle of the cyclic voltammogram obtained at 25 mV/s for the membranes systems studied. All tests were made in the presence of 1 mL of SBF.

**Figure 6 nanomaterials-13-01493-f006:**
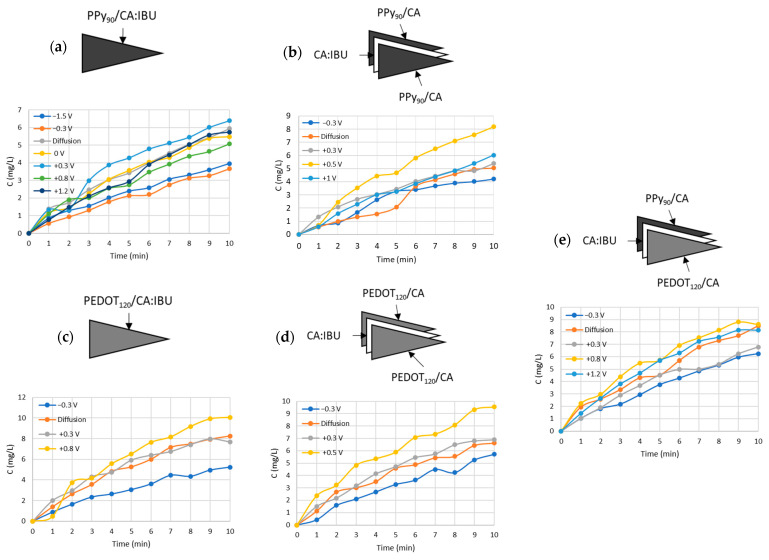
Influence of electrical stimulus on the drug release profile of ibuprofen for: (**a**) PPy_90_ Single System; (**b**) PPy_90_ Rim System; (**c**) PEDOT_120_ Single System; (**d**) PEDOT_120_ Rim System; (**e**) Mix Rim System.

**Figure 7 nanomaterials-13-01493-f007:**
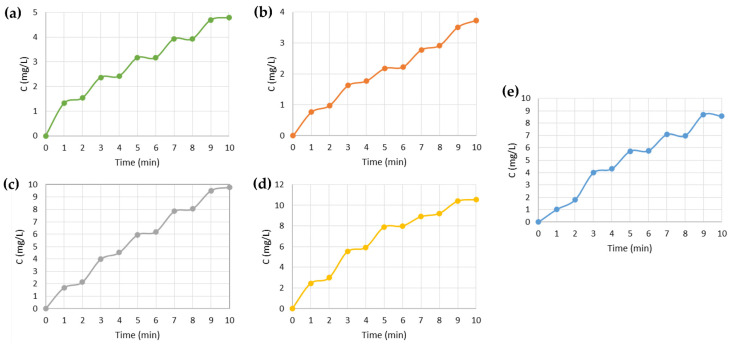
Release profiles of the membrane systems studied: (**a**) PPy_90_ Rim System; (**b**) PPy_90_ Single System; (**c**) PEDOT_120_ Rim System; (**d**) PEDOT_120_ Single System; (**e**) Mix Rim System.

**Figure 8 nanomaterials-13-01493-f008:**
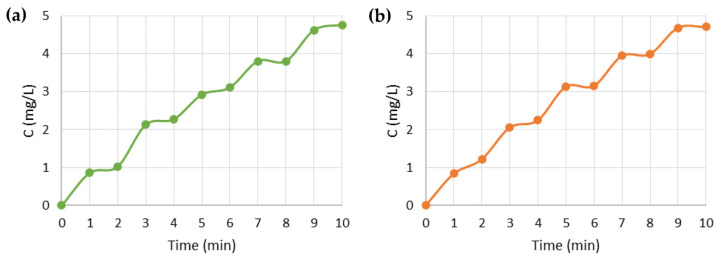
Drug release profiles applying voltage with the battery for: (**a**) PPy_90_ Rim System prototype; (**b**) PPy_90_ Single System prototype.

**Table 1 nanomaterials-13-01493-t001:** Voltages to be intercalated in each system and obtain an “ON/OFF” pattern.

Systems	OFF (V)	ON (V)
PPy_90_ Rim System	−0.3	+0.5
PPy_90_ Single System	−0.3	+0.3
PEDOT_120_ Rim System	−0.3	+0.5
PEDOT_120_ Single System	−0.3	+0.8
Mix Rim System	−0.3	+0.8

**Table 2 nanomaterials-13-01493-t002:** Concentration and percentage of ibuprofen in samples analysed by UPLC-MS/MS.

Systems	Voltage (V)	IBU (µg/L)	IBU (%)
PPy_90_ Rim System	+0.5	12	0.38
	+1.0	<LOD	--------
PPy_90_ Single System	−0.3	<LOD	--------
	Diffusion	2.3	0.074
	0	24	0.77
	+0.3	33	1.0
	+0.8	19	0.61
	+1.2	14	0.45
PEDOT_120_ Single System	+0.3	12	0.38
	+0.8	24	0.77
Mix Rim System	+0.8	25	0.80
	+1.2	16	0.51

## Data Availability

The data presented in this study are available upon request from the corresponding authors.

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
