# Peer review of "Functionalisation of Electrospun Cellulose Acetate Membranes with PEDOT and PPy for Electronic Controlled Drug Release"

_nanomaterials, 2023, doi:10.3390/nano13091493_

Round 1

Reviewer 1 Report

The paper presents an interesting concept of controlled drug delivery system.

The fundamental question, which should be discussed and presented to the reader is the applicability of such system. The example of application should be described very clearly. The on/off scheme for drug administration in the case of topical drug delivery systems/patches is not convincing, because the easiest way to stop the drug release is the removing of the patch from the application place. The Authors have to rethink the possible application of such system.

Another issue that should be addressed in reviewed version of the paper is the use of ibuprofen as a model drug.  However, it is clear that it is only the model, its application from the point of view of topical application needs justification.

Minor issues:

Please check the explanation of abbreviations e.g. IBU in abstract.

Author Response

The answer to the reviewer is in the attachment.

Reviewer 2 Report

The present manuscript is somewhat an extension of the group’s previous work performed on the topic of potential controlled release of ibuprofen (IBU) loaded in cellulose acetate fibers (DOI: 10.1039/D1TB00249J) in the presence of PPy. Although some of the elements are returning elements, in the current study the influence of PEDOT and some novel membrane assemblies were also tested. However, in the light of previous work, it was expected a more rigorous and systematic study of the mentioned systems. The final aim and applicability of an electronically controlled IBU delivery system with such a short ON/OFF sequences has not been justified. The manuscript is not well structured, results and discussions are very fuzzy and difficult to follow, the unrefined methodology (role and value of the LC-MS/MS analysis has not been fully explored; inappropriate use of electrochemical setup and data interpretation), data is not throughout consistent, conclusions fuzzy and potentially misleading (i.e. still not clear what is the %IBU released from the membranes), no statistical treatment of the data is provided, and many more.

 In light of the above statements and some of the additional justifications presented below, the present manuscript is not suitable for publication in its current form. After major rethinking of the experimental protocol, additional experiments performed and a more rational data interpretation it might be resubmitted for evaluation.

Additional major and minor issues related to the study:

1.       The topic of the present study does not necessarily aligned with the set aims and objectives of the journal.

2.       Introducing novel components to the transdermal drug delivery system it is expected to cover some aspects on biocompatibility, lack of immune reactions on the long (repetitive use) and short term, cytotoxicity, etc. Former studies on the lack of cytotoxic effects of PEDOT derivatives (PEDOT-TUDCA) does not represent a proof for novel derivatives. As it turned out there is a major issue of leaching residual reagents, degradation and or by-products from synthesis, thus their potential impact on the health should be mentioned, discussed and potentially explored (given the fact that such a highly sensitive and selective method of analysis (UPLC-MS/MS) is at the hand of the authors).

3.       Section 2.6 and 3.3 – It has not been discussed the justification of selecting a 2-electrode configuration. How was the potential drops at both electrode/electrolyte interfaces controlled in the 2-electrode system? Unfortunately, such voltamogramms are impossible to interpret and more even so to compare on different systems. Not clear what did authors pursue in showing the influence of scan rate on the 3rd cycle of the voltammogram? Lines 374 – 375: as previously stated, the reported values of potential are not relevant and probably may not be matched by the application of an external potential on the drug loaded membrane using a battery. The interpretations provided in this section are highly tendentious and quite unreliable. Line 391: it is not only the conductance of the drug loaded polymer is important in forecasting the efficacy of electrostimuli-based drug release, but also the inherent redox activity of the polymer itself.

4.       It is not clear how drug load of the membranes has been assessed and content uniformity ensured between different tested membranes? Especially as towards the end (line 587) this is mentioned as one of the potential sources of variability.

5.       The long discussions on developing and exploiting the spectrophotometric assay for the monitoring of IBU is completely redundant as it has been shown to be prone to serious interference and overestimations. All assays should be performed with a more selective method, however only some samples (not even clear which) has been finally analyzed UPLC-MS/MS; although interpretations were being done on both sets of data.

6.       Section 3.4: How were compensated the systems for the changes at different time intervals of the passive diffusion?

7.       Figure 5: no statistical evaluation demonstrating variability and statistical significance

8.       Lines 436-438: How would this hypothesis explain the correlation of IBU release upon PPy polaron changes when held inbetween two identically changed (IBU repulsive) PPy membranes (in case of the Rim system) in comparison with the PPy Single system?

9.       Lines 442-483: very unclear discussions. It is not clear why a higher potential (i.e. +0.8V or +1V) would reverse the IBU release when compared to +0.5V? In principle at this point no changes in the charge of the polymeric matrix should occur.

10.   Lines 460-461: Inconsistencies in reasoning during experimental design and data interpretation when discussing the selection of applied potential values for the PPy and PEDOT systems.

11.   Lines 495: The required potential value should not be related to the number of existing active sites.

12.   Figure 6: These values should be comparatively presented when no application of potential is carried out. Additionally statistical treatment is also required (replicate analysis).

13.   Figure 7: It is difficult to understand how the real time monitoring of the released IBU levels has been done. How reproducible is the control of IBU release?

14.   Lines 541-542: Calculations of drug load seem to be unclear.

15.   Lines 549: Minor release of byproducts are not able to significantly affect IBU absorbance, unless a high proportion is release of them or a certain chemical interaction is also ongoing. These so-called “minor byproducts” need to be identified.

16.   Given the issues of selectivity, all assays should be made by LC-MS/MS analysis and only data resulting from reliable analytical methods should be processed and interpreted.

17.   Table 2: To what setup exactly correspond the IBU levels and %IBU released presented in the table?

18.   Confusing and fuzzy discussions on Raman analysis.

19.   Line 593: Such assumptions are not founded considering the weak Raman signals at the beginning

20.   Conclusions: Conclusions should not be a short version of the Results and discussions.

Author Response

(The authors gave the same response as above.)

Reviewer 3 Report

The authors present an interesting application of PEDOT and PPy for electronic controlled drug release, that would allow to modulate the release to wounds.

The paper is well written (minor improvements in English required), and the subject is interesting. The experimental design would be appropriate, even if the PPy Single system contains no drug before the release tests.

The paper can be published after minor revisions:

a) The sketch of the prototype should be inserted in section 2.8 (it actually appears later in Figure 7).

b) Figure 3: in some of the images insets are present, with enlargements of details, but no bar scale/enlargement is given. Also, clarify what is the number in the blue box in the upper rigth corner.

c) In Figure 4 for PPy90 Single cathodic and anionic peaks described are not seen.

Author Response

(The authors gave the same response as above.)

Reviewer 4 Report

The manuscript "Functionalization of cellulose acetate membranes with PEDOT and PPy for electronic controlled drug release" by Lago et al. reports the polypyrrole and poly(3,4-ethylenedioxythiophene) functionalization of electrospun cellulose acetate membrane encapsulated with ibuprofen molecules. The prepared membranes are characterized and release profiles upon very low electrical potentials are presented. The paper could be interesting to the readers, but before to be published some clarifications are mandatory. Following are comments/suggestions for improvement:

- did the authors quantify experimentally the IBU content in each prepared sample? How much IBU has been embedded in cellulose acetate membranes functionalized and non-functionalized? The only released IBU has been quantified. Has been released the whole amount of IBU from the prepared systems?

- why the authors did not use the same electrical stimulus values when the influence of this parameter on the ibuprofen release profile for each sample has been studied (Figure 5)?

- Could the authors propose a mechanism of IBU release from prepared membrane systems? Where IBU molecules are localized through membrane? How IBU is bonded to the prepared materials membrane? And how upon the applied voltage the IBU leaves the system?

- the release profiles are given for only 10 minutes. Is there any reason for considering this time? If more IBU will be embedded in the membrane, the same time will be available for release experiments? If off cycle will be a longer than that considered in the manuscript, will be affected the release of IBU from the prepared membranes?

Author Response

(The authors gave the same response as above.)

Reviewer 5 Report

The authors have presented their studies on the electrically modulated release of a model drug from cellulose acetate membranes functionalized with PEDOT and Polypyrrole. The journal Nanomaterials focusses on studies on materials with a dimension in the nanomaterial scale (less than 100 nm), however as evident from Figure 3, the fibers exceed the nanomaterial size classification. They are larger than 100nm so this manuscript would better suit publication in Materials or Polymers.

However, the following should be addressed before publication:

1. Introduction - please specify clearly the novelty of this study.

2. Section 2.2. What effect does the washing regime have on the drug content in the membranes?

3. Section 2.7. In drug release studies, is the extracted media returned to the release vessel or is it replaced with fresh media?

4. Section 3.5. Drug release in the range 300-800% have been observed and this has been attributed to other substances released in the media. Are monomers being released in the media? If yes, then the polymerization regime or the washing regime should be revisited to eliminate them.

Author Response

(The authors gave the same response as above.)

Round 2

Reviewer 1 Report

I have found the answers to my questions satisfying.

Author Response

Dear reviewer, 

Thank you for your kind answer and suggestions to improve the quality of the manuscript.

Kind regards,

All the authors

Reviewer 5 Report

The authors have made significant improvements to the manuscript and have attended to the reviewers' queries/ comments in a satisfactory manner. This manuscript may now be considered for publication, albeit some minor attention to the English language.

Author Response

(The authors gave the same response as above.)
